# CABA: A Collusive Aggregation-Emergent Backdoor Attack in Federated Learning

## Abstract

Federated Learning (FL) has been shown to be vulnerable to backdoor attacks conducted by malicious clients. Although many studies have enhanced the stealthiness and durability of backdoors, the full potential of collusive attacks in FL remains underexplored. Existing collusive attacks typically adopt a strategy where each malicious client trains independently. These attacks inevitably embed backdoor features into the uploaded updates and make them susceptible to detection. To fully exploit the collaborative capabilities of malicious clients, we propose a novel collusive attack, named CABA (Collusive Aggregation-based Backdoor Attack), where the backdoor behavior emerges only during model aggregation. In CABA, multiple malicious clients jointly craft a set of updates that individually exhibit no backdoor characteristics, allowing them to bypass defense mechanisms. However, when aggregated, these updates manifest the backdoor in the global model. Extensive experiments demonstrate that our proposed attack can successfully bypass six state-of-the-art defense mechanisms, demonstrating superior stealth and attack efficacy compared to existing collusive approaches. Our research highlights the critical importance of developing defense mechanisms that can inspect the combined behavior of model updates after aggregation.

## 1 Introduction

Federated Learning (FL) (Yang et al., 2019; Kairouz & et al., 2021) enables collaborative model training across numerous clients under the coordination of a central server, epitomized by algorithms like FedAvg (McMahan et al., 2017). This paradigm offers significant privacy advantages by allowing clients to keep their raw data local. However, the opacity of local training processes exposes FL to security threats, notably *Byzantine attacks* (Deshmukh, 2024; Li et al., 2024). In these attacks, malicious clients submit arbitrary model updates to corrupt the global model or impede convergence. Among these threats, *Backdoor attacks* (Mothukuri et al., 2021; Fang & Chen, 2023) are particularly insidious. In this scenario, an adversary implants a hidden backdoor into the global model, causing it to misclassify inputs embedded with a specific trigger—for instance, classifying any image of a "dog" as a "cat" whenever a small green square is present in the corner—while maintaining high accuracy on benign data. Previous work (Bagdasaryan et al., 2020) has shown that such attacks can be mounted by a single malicious client simply submitting a poisoned model update.

To enhance the stealth and effectiveness of these attacks, recent research has shifted towards *collusive strategies* (Xie et al., 2020; Lyu et al., 2025; Li et al., 2023) that leverage multiple malicious clients. These strategies represent a clear escalation, moving beyond single-point failures. For example, attacks like CoBA (Lyu et al., 2025) coordinate multiple attackers to jointly optimize a more subtle backdoor trigger and their corresponding local models, making the attack more potent. Other advanced methods like 3DFed (Li et al., 2023) employ a "decoy" strategy, where some malicious clients submit crafted updates to manipulate the server's statistical baseline, thereby helping to camouflage the primary malicious updates from their collaborators. Despite their sophistication, these approaches still operate under a common, limiting paradigm: **each malicious client independently trains and submits a complete, functional backdoor model.** This represents a relatively shallow form of collusion, where collaboration is used for concealment rather than fundamentally altering the nature of the attack itself.

This independent training paradigm, even when coordinated, creates a critical vulnerability. Because each malicious update contains the full statistical and structural signature of a backdoor, it remains detectable by robust defense mechanisms(Blanchard et al., 2017; Deshmukh, 2024; Fang et al., 2025). Current defenses primarily operate at the server by inspecting incoming model updates. One class of defense relies on *statistical anomaly detection* (Yin et al., 2018; Nguyen et al., 2022); for instance, Multi-Krum (Blanchard et al., 2017) identifies outliers based on Euclidean distance between model updates, while Foolsgold (Fung et al., 2020) flags colluding clients by detecting abnormally high cosine similarity in their gradient updates. A second, more powerful class of defenses (Li & Dai, 2024; Xu et al., 2025; Rieger et al., 2022) inspect the *intrinsic properties* of a single model. Proactive defenses like BackdoorIndicator (Li & Dai, 2024) investigates the interaction of sequentially inserted backdoors by pre-embedding a backdoor into the global model using out-of-distribution (OOD) data, and it classifies a client as malicious if the client's uploaded update yields high accuracy on this pre-embedded backdoor. Consequently, while a collusive attack with decoys might bypass simple distance-based or PCA-based checks, its core backdoor update can still be identified and rejected by these more advanced, intrinsic-aware defenses.

**Our contribution.** Through our survey of current research on collusive attacks and defenses, we have found that existing collusive attacks are insufficient to bypass state-of-the-art defense techniques. The full potential of collusive mechanisms remains underexplored. To fully explore the upper bounds of collusive threats, we design and propose a novel **C**ollusive **A**ggregation-emergent **B**ackdoor **A**ttack, named **CABA**. Instead of each attacker contributing a complete backdoor model, CABA distributes the training across multiple clients. In CABA, the adversary constructs a set of malicious updates via joint backdoor training among multiple malicious clients. Individually, each client's update is merely a benign-looking fragment; the complete backdoor functionality only manifests after the server aggregates these distinct partial updates. This fragmentation ensures that no single update possesses the statistical anomalies or the complete intrinsic properties of a backdoor, allowing it to bypass both anomaly detection and property-based defenses. Extensive experiments demonstrate that CABA bypasses six state-of-the-art defense mechanisms, proving to be both stealthier and more effective than existing collusive attacks. Our research exposes a blind spot in existing defense mechanisms and demonstrates the importance of validating the combined model after aggregation.

## 2 PRELIMINARIES

### 2.1 FEDERATED LEARNING

In the traditional horizontal FL paradigm, there is a central server $S$ responsible for orchestrating all clients and aggregating local model updates from clients into the global model. In each training round $t$, the server first broadcasts the global model $G^t$ to the selected subset $C^t$ of clients. Each selected client $i \in C^t$ trains its local model $L_i^t$ locally using its own dataset $D_i$. Once Client $i$ finishes training, it only uploads the local model update $g_i^t = L_i^t - G^t$ instead of data to the server to keep data privacy. After receiving these model updates, the server aggregates these updates into the global model according to the aggregation algorithm. FedAvg (McMahan et al., 2017) is the baseline aggregation algorithm. In FedAvg, the server computes $G^{t+1} = G^t + \frac{\eta}{|C^t|} \sum_{i \in C^t} g_i^t$ as the new aggregated global model for the next round, where $\eta$ is the learning rate and $|C^t|$ is the number of selected clients in the current round. The server repeats the above training process until the global model converges. In the rest of our paper, we use FedAvg as our default aggregation algorithm.

### 2.2 BACKDOOR ATTACKS IN FL

Backdoor attacks aim to implant the backdoor into the model by injecting training samples patched with specified triggers and corresponding target labels during the model training process. The backdoored model will output the target label designated by the attacker once the model meets the testing samples containing the trigger, while it will perform normally under other circumstances. The distributed architecture and non-transparent local training process of FL present a viable attack surface for backdoor attacks. The adversary can simply participate in the FL training process and upload the backdoored local model to the server to introduce the backdoor functionality into the global model. The Model Replacement Attack (Bagdasaryan et al., 2020) points out this vulnerability and success-

fully backdoors FL through replacing the uploaded update with the backdoored model update $g_b^t$ trained by the adversarial loss $\mathcal{L}_{adv} = \alpha \mathcal{L}_{class} + (1 - \alpha)\mathcal{L}_{ano}$ on the partially backdoored dataset $D_{backdoor}$. It designs the normalization term $\mathcal{L}_{ano}$ and scales the backdoor updates by the factor $\gamma$ to evade the anomaly detection and ensure the effectiveness of the replacement. Based on this vanilla method, there are also many advanced methods proposed to improve the stealth and durability of the backdoor attack in FL (Zhang et al., 2022; Dai & Li, 2023; Krauß et al., 2024).

## 2.3 ROBUST AGGREGATION ALGORITHMS FOR FL

In FL, robust aggregation algorithms are designed to protect the central model from poisoning or Byzantine attacks by identifying and filtering out malicious model updates from adversarial clients during the aggregation phase. While conventional Federated Averaging is highly vulnerable to such attacks, the objective of robust aggregation is to ensure that the aggregated global model update promotes convergence, even in the presence of a subset of malicious clients. Formally, a robust aggregation algorithm Robust_AGG() aims to compute an aggregated update such that the new global model $G^{t+1} = \text{Robust\_AGG}(\{g_k^t\}_{k \in C^t})$ approximates the ideal model derived solely from honest clients. A range of robust aggregation algorithms exists to secure FL systems. Multi-Krum (Blanchard et al., 2017) (Distance-Based) identifies malicious clients by calculating a score based on the distance of each update to its neighbors and discards those that are statistical outliers with the highest scores. FLAME (Nguyen et al., 2022) (Clustering-Based) uses HDBSCAN to group model updates based on cosine similarity, then identifies the largest and most stable cluster as benign while rejecting smaller clusters and outliers as potentially malicious. FoolsGold (Fung et al., 2020) (Similarity-Based) penalizes coordinated malicious clients by down-weighting the influence of updates that are highly similar to many others. RFLBAT (Wang et al., 2022) (Projection-Based) identifies and removes backdoor attacks by using PCA and K-means clustering to detect client updates with unusually large parameter changes, then excludes these outlier contributions. DeepSight (Rieger et al., 2022) (Behavior-Based) detects malicious participants by analyzing model update patterns—examining prediction differences, neuron update magnitudes, and data distribution homogeneity—to group and reject clients showing backdoor-like training behaviors. BackdoorIndicator (Li & Dai, 2024) (Property-Based) is a proactive defense mechanism where the central server in FL injects a temporary indicator task, constructed from out-of-distribution (OOD) data, into the global model and then identifies malicious updates by observing the preservation of this task's accuracy, a side-effect intrinsically caused by the OOD nature of the backdoor training process itself.

## 3 PROBLEM DEFINITION

### 3.1 THREAT MODEL

**Attacker's Goal.** The attacker's objective is to embed backdoors into the global FL model via a subset of compromised clients, while preserving the model's performance on its main task. Specifically, the backdoored global model should exhibit the backdoor behavior designated by the attacker while performing as well as the benign classifier when encountering the original samples without the trigger. Moreover, the attacker seeks to conduct the backdoor injection covertly, thereby circumventing server-side defenses.

**Attacker's Knowledge and Capability.** The attacker is assumed to control a subset of clients, granting full authority to modify their local data, training procedures, and submitted model updates; these compromised clients may also collude. However, the attacker cannot access the central server or benign clients, nor eavesdrop on their communications. We evaluate this threat under two knowledge settings. In a white-box scenario, the attacker is omniscient—possessing knowledge of all client data distributions and server-side mechanisms—to establish the attack's upper bound. Conversely, in a realistic black-box scenario, the attacker's knowledge is restricted to its own compromised clients, consistent with standard FL protocols.

### 3.2 PROBLEM FORMULATION

**System Model.** Our paper considers a horizontal FL system containing $N$ clients. Each client $i$ owns a local dataset $D_i = \{(x_j^i, y_j^i)\}_{j=1}^{d_i}$ where $d_i = |D_i|$ and $(x_j^i, y_j^i)$ represents the feature vector

and ground-truth label of the $j$-th data sample in $i$-th client's dataset. The data can be distributed in either an IID (Independent and Identically Distributed) or non-IID manner across clients. In each FL training round, the central server randomly selects a subset $C^t$ containing $N_s$ out of $N$ clients to participate in the global model aggregation process. In each joint training round $t$, as illustrated in Section 2.1, the server broadcasts the c and receives local updates $\{g_k^t\}_{k \in C^t}$ from a selected subset $C^t$ of clients. Different from FedAvg, we assume that the server deploys the defense mechanism against adversarial attacks during the aggregation phase. The server aggregates all local updates according to the aggregation algorithm with the defense mechanism, which is denoted as the function Robust_AGG(). At the end of this training round, the global model $G^t$ is transformed to $G^t$ for the next round $t + 1$ by the formula: $G^{t+1} = \text{Robust\_AGG}(\{g_k^t\}_{k \in C^t})$

**Attack Objective.** To conduct collusive backdoor attacks, we assume the attacker controls a set of multiple clients $C_{adv}^t \subset C^t$ in each round $t$ by default. We denote the backdoor trigger as $x_{tri}$ and the target label as $\hat{y}$. The attack objective of the collusive backdoor attack in each round $t$ can be formulated as the following:

$$x_{tri}^*, \{g_m^t\}_{m \in C_{adv}^t}^* = \underset{x_{tri}, \{g_m^t\}_{m \in C_{adv}^t}}{\arg\min} \underbrace{\sum_{j=0}^{|D_{test}|} \ell_{G^{t+1}}(x_j, y_j)}_{main\ task\ loss} + \underbrace{\sum_{j=0}^{|D_{test}|} \ell_{G^{t+1}}(x_j + x_{tri}, \hat{y})}_{backdoor\ loss} \quad (1)$$

where $\ell_{G^{t+1}}$ denotes the classification loss function given the labeled data samples in the test dataset. The first term in the objective function guarantees the accuracy of the main classification task of FL and the second term ensures the effectiveness of the backdoor injection. Therefore, the primary challenge of designing the collusive backdoor attack is a distributed optimization problem to find a solution of $x_{tri}, \{g_m^t\}_{m \in C_{adv}^t}$ to minimize both the first loss term and second loss term at the same time against the robust aggregation algorithm.

## 4 OUR COLLUSIVE ATTACK: CABA

### 4.1 MOTIVATION

While recent studies explore collusive backdoor attacks, their methods still concentrate the core backdoor training, creating a critical vulnerability. Approaches like DBA and CoBA produce malicious updates that are highly similar to each other, making them easily detectable by clustering or similarity-based defenses. As illustrated in Figure 1, malicious updates are homogeneous and distinct from benign updates. Other methods, such as 3DFed, obfuscate statistics but still confine the actual backdoor injection to a single client's update. This centralization ensures that intrinsic backdoor features persist within that individual update, rendering it vulnerable to property-based defenses like BackdoorIndicator. The fundamental limitation across these attacks is their failure to distribute the backdoor training process itself. In this paper, we address this gap by distributing the backdoor training across multiple colluding clients. This strategy aims to ensure that no single malicious update exhibits the complete, detectable characteristics of a backdoor, thus significantly enhancing the attack's stealth.

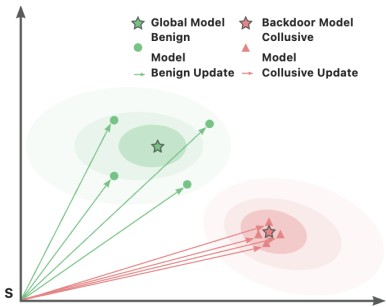

Figure 1: The visualization of previous collusive attacks' updates.

### 4.2 OVERVIEW OF CABA

Our collusive backdoor attack, CABA, is based on the rationale previously discussed in Section 4.1. The core strategy involves multiple clients who jointly train a unified backdoor model and then partition it. Each colluding client uploads only its assigned sub-model. This distribution of the backdoor's features ensures that each individual update remains stealthy, exhibiting minimal deviation from benign models and thus being harder to detect than a conventional single-client attack. We designed the CABA algorithm to implement this powerful collusive attack, and its overall pipeline is depicted in Figure 2.

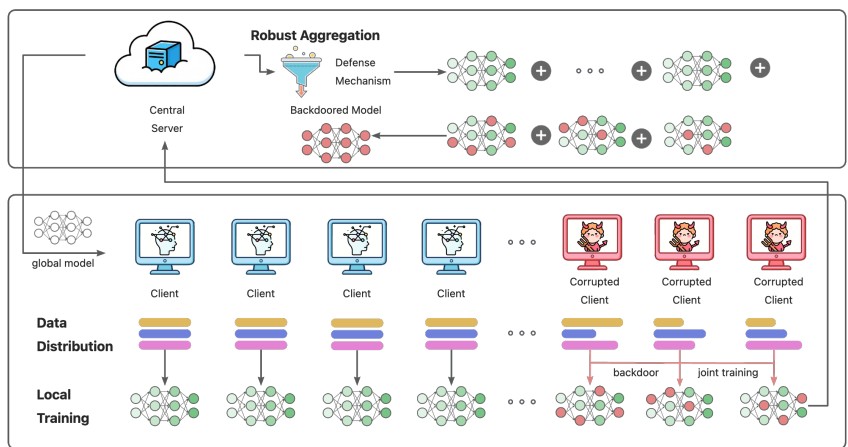

Figure 2: The Overview of CABA.

Our proposed CABA framework constructs stealthy backdoor updates in FL through a four-phase process per round. First, in the *Data Redistribution* phase, attackers engineer a Non-IID data distribution among compromised clients to foster model diversity. This is crucial for the subsequent *Benign Model Pre-training* phase, where distinct benign models are trained on these skewed datasets, serving as initializations. The third phase, *Joint Trigger Tuning*, optimizes the trigger pattern itself to enhance attack efficacy. In the final and pivotal *Joint Backdoor Training* phase, the benign models are collectively fine-tuned on a poisoned dataset using a composite loss function. This function is designed to enforce the backdoor objective only on the aggregated model while maintaining the benign appearance of individual sub-models to evade defenses. Finally, the compromised clients upload these individually innocuous sub-models, with the backdoor activating only upon aggregation by the central server. Figure 3 visualizes the effect of the collusive malicious updates generated by the four-phase process described above. As shown, the malicious updates are indistinguishable from the benign updates.

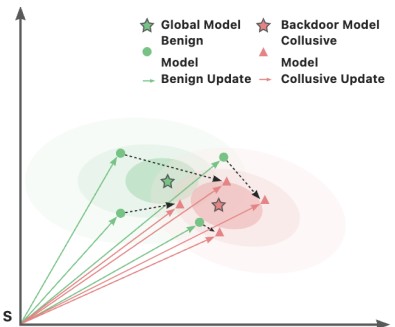

Figure 3: The visualization of our CABA's malicious updates.

### 4.3 DETAILED METHODOLOGY

Algorithm 1 details the procedure of our CABA algorithm for malicious clients to jointly train the optimal trigger and their colluding models. We now detail each stage of the algorithm.

**Data Redistribution.** Model homogeneity, resulting from the standard IID data assumption in FL, is a primary obstacle to our attack. With sub-models converging to near-identical parameters (cosine similarity ¿ 0.99), the aggregated model effectively mirrors the sub-models. This makes it impossible to simultaneously achieve the conflicting objectives of individual model stealth and aggregate backdoor efficacy. Our joint training method, which distributes backdoor functionality, thus requires initial model diversity, necessitating heterogeneous starting points that an IID setting cannot provide.

To ensure the training of diverse benign models, we implement a data redistribution phase to engineer a Non-IID partition. First, we aggregate data from all malicious clients $C_{adv}^t$ into a unified dataset $D_{sum}^t = \bigcup_{k \in C_{adv}^t} D_k$. We then employ a Dirichlet distribution, controlled by a parameter $\lambda$, to sample a class probability vector for each client, which dictates the allocation of data from $D_{sum}^t$. A smaller $\lambda$ induces a more skewed, Non-IID distribution. The redistribution is formulated

---

**Algorithm 1:** CABA

---

1  **AdversaryExecutes:** (on behalf of compromised clients $C_{adv}^t$)
2      *// Data Redistribution*
3      Pool and redistribute datasets among all compromised clients based on Equation 2;
4      *// Benign Pre-training and Trigger Tuning*
5      **for** *each client $k \in C_{adv}^t$* **do**
6          Pre-train a benign base model $H_k$ on its redistributed dataset based on Equation 3;
7      **end**
8      *// Joint Trigger Tuning*
9      Optimize the backdoor trigger $x_{tri}$ using the pooled dataset based on Equation 4;
10     *// Joint Backdoor Model Training*
11     Aggregate current client attack models to get the aggregated model $G_{agg}$;
12     **for** *each data batch in pooled dataset* **do**
13         Form a poisoned data batch using the trigger $x_{tri}$;
14         Jointly update all client models $\{G_k\}$ based on a composite loss based on Equation 5
15     **end**
16     Return the set of malicious models $\{G_k\}_{k \in C_{adv}^t}$ to the server;

---

as:

$$\widetilde{D_k} = \{(x,y) \in D_{\text{sum}}^t \mid y = m \text{ with probability } p_{k,m}\} \tag{2}$$

Each colluding malicious client replaces its local dataset $D_k$ with $\widetilde{D_k}$ to be used for the subsequent joint backdoor training.

**Benign Model Pre-training.** To create the necessary model diversity for joint training, we introduce a Benign Model Pre-training phase. Each malicious client trains the global model $G^t$ on its unique, redistributed dataset $\widetilde{D_k}$, adhering to the standard FL protocol. This process yields a set of distinct benign models $\{G_k^t\}$ that serve as heterogeneous starting points for the subsequent joint backdoor fine-tuning. The training follows the standard gradient descent update:

$$G_k^t \leftarrow G_k^t - \eta \sum_{(x_j, y_j) \in \widetilde{D_k}} \nabla \ell_{G_k^t}(x_j, y_j) \tag{3}$$

where $\ell_{G_k^t}$ is the cross-entropy loss. To maintain stealth, training hyperparameters, such as learning rate and iterations, mirror those of honest clients. The outcome is a unique benign model $G_k^t$ for each attacker, ready for the next phase.

**Joint Trigger Tuning.** To enhance stealth and reduce the conflict between the main and backdoor tasks, we implement a Joint Trigger Tuning phase. In this phase, we treat the trigger $x_{tri}$ as an optimizable variable. We use SGD to simultaneously update both the trigger and the global model $G^t$ on the aggregated malicious dataset $D_{sum}^t$. The model is trained on samples patched with the trigger but retaining their ground-truth labels, denoted as $(X + x_{tri} \cdot \mathbf{1}_{|X|}, Y)$. To constrain the trigger's modifications, a projection operation Proj is applied after each update, projecting the trigger into an $L2$-norm hypersphere of radius $N$. The optimization process is formulated as:

$$x_{tri} \leftarrow x_{tri} - \text{Proj}(\eta \nabla \ell_{G^t}(X + x_{tri} \cdot \mathbf{1}_{|X|}, Y), N, 2) \tag{4}$$

This approach finds an optimal trigger that minimally impacts main task accuracy, thereby aligning the objectives of the two tasks. While we use a pixel pattern as our default example, this method is trigger-agnostic.

**Joint Backdoor Training.** Joint backdoor training aims to create a set of malicious sub-models that seem benign individually but manifest a backdoor when aggregated. This is achieved by optimizing a composite loss function, $\ell_{joint}$. First, to instill the backdoor in the aggregated model of malicious clients, which is formulated as $G_{agg}^t \leftarrow \sum_{k \in C_{adv}^t} \frac{1}{|C_{adv}^t|} \cdot G_k^t$, a backdoor loss, $\ell_{G_{agg}^t}(X_b, Y_b)$, is applied to it. Simultaneously, to make each individual sub-model appear benign, an opposing loss, $\sum_{k \in C_{adv}^t} \ell_{G_k^t}(\widehat{X_k}, Y_k)$, forces each one to predict the correct ground-truth label on data with the backdoor trigger. To evade server-side defenses, two regularization terms are added:

$\ell_{cos}(\{G_k^t\}_{k \in C_{adv}^t})$ minimizes the cosine similarity between the sub-models' updates, and a term based on $||G_k^t - H_k^t||$ restricts the L2-norm deviation of each sub-model from its initial state. The final combined loss function is a weighted sum of these components:

$$\ell_{joint} = (1-\alpha) \sum_{k \in C_{adv}^t} \ell_{G_k^t}(\widehat{X_k}, Y_k) + \alpha \ell_{G_{agg}^t}(X_b, Y_b) + \beta \ell_{cos}(\{G_k^t\}_{k \in C_{adv}^t}) + \gamma \sum_{k \in C_{adv}^t} ||G_k^t - H_k^t|| \quad (5)$$

This allows for the simultaneous training of all sub-models, using a custom structure to backprop-agate gradients from the aggregated model to the individual ones, ultimately producing seemingly benign models that can collusively inject a backdoor.

## 5 EVALUATION

### 5.1 EXPERIMENTAL SETUP

**FL settings.** We conduct evaluations on the **CIFAR-10** dataset. The model employed for all experiments is **ResNet-18**. Following standard FL benchmarks, we set the total number of clients to 100, with 4 clients being compromised by default, over 500 training rounds. In each round, 20 clients are randomly sampled to participate and all compromised clients are always being selected. To simulate a non-IID data distribution, we partition the dataset among clients using a Dirichlet distribution (Li et al., 2022) with a factor of 0.5. Each client trains its local model using SGD with a learning rate of 0.1 and a batch size of 64 for 5 local training rounds. The server uses the FedAvg algorithm for aggregation.

**Baseline attacks and Defenses.** To provide a comprehensive comparison, we evaluate against four state-of-the-art baseline collusive backdoor attacks mentioned above: **DBA** (Xie et al., 2020), **Chameleon** (Dai & Li, 2023), **CoBA** (Lyu et al., 2025), and **3DFed** (Li et al., 2023). We evaluate the robustness of our attack against a suite of seven defenses: **DeepSight** (Rieger et al., 2022), **FLAME** (Nguyen et al., 2022), **FoolsGold** (Fung et al., 2020), **BackdoorIndicator** (Li & Dai, 2024), **Multi-Krum** (Blanchard et al., 2017), **RFLBAT** (Wang et al., 2022), and we also consider FedAvg as a **NoDefense** baseline.

**Metrics.** We use three metrics for evaluation: **Accuracy (Acc, %)**, which measures the model's performance on clean test samples. **Attack Success Rate (ASR, %)**, which measures the model's prediction accuracy on samples embedded with the backdoor trigger, targeting the specified class. **Defense Bypass Rate (DBR, %)**, which quantifies the percentage of malicious updates that are not detected or filtered by the defense mechanism.

**Others.** For all baseline attacks and defenses, we adhere to the hyperparameter settings specified in their original papers. All experiments are conducted five times using different random seeds, and we report the average results. All experiments were conducted on a single NVIDIA A800 GPU.

### 5.2 EXPERIMENTAL RESULTS

#### 5.2.1 ATTACK PERFORMANCE COMPARISON

To evaluate the efficacy of our proposed CABA attack, we conducted a comprehensive set of experiments comparing it against several state-of-the-art backdoor attacks—DBA, Chameleon, CoBA, and 3Dfed—in the face of various robust aggregation defenses. The experiments were performed on the CIFAR-10 dataset under both IID and Non-IID data settings, with attacks initiated at different training rounds (400, 800, and 1200). As shown in Table 1 and Table 3 in Appendix, in the absence of any defense (Nodefense), most attacks achieve a high ASR, with CoBA and our CABA consistently reaching near-perfect ASRs. However, when defenses such as Multi-Krum, FLAME, and Foolsgold are deployed, the effectiveness of existing attacks like DBA and Chameleon is significantly diminished, as indicated by their substantially reduced ASRs. In contrast, CABA maintains a high ASR, often exceeding 90%, across almost all defenses and start rounds in both IID and Non-IID scenarios. This demonstrates CABA's superior stealth and robustness. Notably, the 3Dfed attack failed against the Deepsight defense in the Non-IID setting; this is attributed to the excessive injection of decoy models, which disrupted the clustering mechanism essential for Deepsight's operation, leading to its failure. The results clearly indicate that CABA is highly effective at bypassing a wide array of ex-

Table 1: Performance comparison of various backdoor attacks against robust aggregation defenses in CIFAR-10 under the IID setting.(Acc%/ASR%)

| Defense | Start Round | DBA | Chameleon | CoBA | 3Dfed | CABA |
|---|---|---|---|---|---|---|
| Nodefense | 400 | 92.30/91.17 | 91.78/30.84 | 92.52/99.99 | 91.66/89.11 | 91.95/92.06 |
| | 800 | 92.51/91.21 | 92.63/30.73 | 92.78/99.99 | 92.47/90.13 | 92.79/92.33 |
| | 1200 | 92.98/92.54 | 92.92/31.46 | 93.04/99.86 | 89.65/90.18 | 92.89/100.00 |
| Multi-Krum | 400 | 89.75/8.62 | 90.06/9.08 | 89.65/100.00 | 89.13/94.84 | 89.45/90.25 |
| | 800 | 92.10/9.56 | 91.62/9.54 | 91.08/99.13 | 91.19/92.84 | 91.81/95.22 |
| | 1200 | 92.70/9.64 | 92.33/9.43 | 92.73/13.09 | 92.10/98.84 | 92.53/98.35 |
| FLAME | 400 | 89.69/92.23 | 88.80/8.52 | 90.40/99.83 | 90.59/8.43 | 90.46/96.78 |
| | 800 | 91.91/92.08 | 91.80/9.38 | 91.81/99.85 | 91.10/9.45 | 92.11/98.35 |
| | 1200 | 92.72/9.80 | 92.93/9.76 | 92.78/12.75 | 92.68/9.65 | 92.58/99.98 |
| Foolsgold | 400 | 91.96/9.95 | 92.19/13.02 | 91.96/11.74 | 92.12/93.97 | 91.60/89.32 |
| | 800 | 92.60/9.99 | 92.51/12.86 | 92.49/13.00 | 92.41/95.37 | 92.52/92.72 |
| | 1200 | 93.07/10.17 | 93.23/12.76 | 93.14/12.33 | 92.81/97.97 | 93.15/99.71 |
| RFLBAT | 400 | 92.02/91.11 | 91.82/31.19 | 92.28/99.80 | 92.10/94.45 | 92.08/90.71 |
| | 800 | 92.44/90.05 | 92.45/35.73 | 92.07/99.96 | 92.43/95.18 | 92.82/93.52 |
| | 1200 | 93.01/88.90 | 92.76/34.38 | 92.85/99.98 | 92.27/96.16 | 93.16/100.00 |
| Deepsight | 400 | 91.39/41.14 | 91.34/9.16 | 91.09/60.76 | 91.21/9.44 | 91.47/44.93 |
| | 800 | 92.25/61.07 | 92.26/9.10 | 92.15/80.06 | 92.30/9.11 | 92.44/48.94 |
| | 1200 | 92.87/75.37 | 89.55/8.35 | 92.79/96.69 | 90.11/9.61 | 93.03/65.15 |
| B.Indicator | 400 | 90.66/36.07 | 91.09/9.58 | 91.72/99.41 | 91.53/30.74 | 92.16/91.50 |
| | 800 | 92.56/71.10 | 92.06/9.27 | 92.85/99.89 | 92.58/20.44 | 92.60/90.03 |
| | 1200 | 92.61/82.39 | 92.40/9.35 | 92.89/99.94 | 92.29/94.86 | 92.68/99.13 |

isting defense mechanisms, showcasing its advanced capabilities in mounting successful backdoor attacks in federated learning.

### 5.2.2 EFFECT OF PARAMETERS

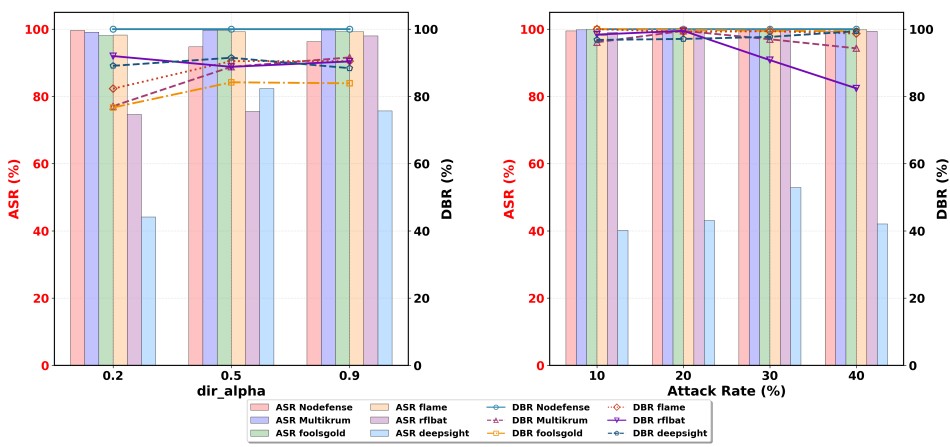

Figure 4: The impact of varying data distributions and the number of malicious clients on the ASR and DBR of the CABA attack.

In this section, we investigate the impact of two critical parameters on our attack's performance: the degree of data non-uniformity across clients, simulated by the Dirichlet distribution parameter `dir_alpha`, and the proportion of malicious clients, referred to as the Attack Rate. The results, illustrated in Figure 4, demonstrate the robustness and efficiency of our proposed method under varying conditions. The left panel of the figure shows the attack's performance as `dir_alpha` varies from 0.2 (highly Non-IID) to 0.9 (approaching IID). The results clearly indicate that our

attack maintains a consistently high Attack Success Rate (ASR), remaining near 100% against the majority of defenses regardless of the level of data heterogeneity. This demonstrates the attack's remarkable robustness to the statistical challenges commonly found in real-world federated learning environments. The right panel of the figure evaluates the attack's efficacy as a function of the Attack Rate, ranging from 10% to 40%. The experiment reveals that our attack is highly efficient, achieving a near-perfect ASR even when only 10% of the clients are malicious. This high performance is sustained as the proportion of attackers increases, underscoring that the attack does not require a large coalition of adversaries to successfully compromise the global model. Collectively, these parameter studies confirm that our attack is potent and resilient across a wide range of challenging and realistic federated learning scenarios.

### 5.2.3 ABLATION STUDY

To validate the contribution of each key component within our proposed CABA framework, we performed a thorough ablation study. The CABA framework is primarily composed of three modules: Data Redistribution (DR), Joint Trigger Tuning (JTT), and Joint Backdoor Training (JBT). We systematically removed each component one at a time and evaluated the attack's performance (ASR) against a suite of six different defense mechanisms. The results, presented in Table 2, reveal the integral role of each module. For instance, without the DR, the ASR against defenses like Foolsgold and FLAME drops dramat-

Table 2: Ablation Study

| Component | w/o DR | w/o JTT | w/o JBT |
|---|---|---|---|
| MultiKrum | 98.01 | 86.43 | 97.22 |
| Foolsgold | 6.41 | 86.33 | 13.61 |
| FLAME | 9.44 | 88.87 | 13.07 |
| Deepsight | 10.82 | 46.37 | 67.10 |
| RFLBAT | 97.27 | 85.47 | 98.78 |
| B.Indicator | 97.21 | 93.92 | 8.44 |

ically to 6.41% and 9.44%, respectively, indicating that DR is critical for evading defenses that rely on update similarity. The removal of JTT also leads to a significant performance decline against most defenses, highlighting its importance in creating a robust backdoor. Similarly, the absence of JBT severely hampers the attack's effectiveness against defenses like Foolsgold, FLAME, and B.Indicator. These results collectively affirm that all three components are essential for the high efficacy of the CABA attack, and they work in concert to bypass a wide range of robust aggregation defenses.

## 6 RELATED WORK

As mentioned in Section 2, backdoor attacks evolved from statistically deviant single-client methods (Bagdasaryan et al., 2020; Sun et al., 2024; Wang et al., 2020) to stealthier collusive strategies that distribute triggers across clients (e.g., DBA (Xie et al., 2020), 3DFed (Li et al., 2023)) to bypass statistical defenses. Consequently, server-centric defenses have advanced from passive, statistical filtering techniques (e.g., Multi-Krum (Blanchard et al., 2017), Foolsgold (Fung et al., 2020), RFLBAT (Wang et al., 2022)), which are vulnerable to adaptive attacks, to robust active methods. A prime example is BackdoorIndicator (Li & Dai, 2024), which actively probes the model with OOD tasks to reliably detect hidden backdoors.

## 7 CONCLUSION

This paper addresses the fundamental vulnerability of existing collusive backdoor attacks in Federated Learning, wherein each adversary submits a complete and individually detectable backdoor model. We propose CABA, a novel "aggregation-emergent" attack where multiple malicious clients collaboratively submit seemingly benign model fragments. The full backdoor functionality manifests only after these fragments are aggregated by the server. Our experiments demonstrate that CABA successfully bypasses six state-of-the-art defense mechanisms, revealing a critical blind spot in current defense strategies that focus solely on inspecting individual client updates. This work underscores the urgent need for future security paradigms to shift towards post-aggregation validation of the global model to counter these emergent threats.

## 8 ETHICS STATEMENT

This paper presents a novel and potent backdoor attack method, CABA, which could potentially be used for malicious purposes. We acknowledge the risks associated with publishing such an attack. However, we strongly believe that the benefits of this research to the security community substantially outweigh the potential harm. The primary purpose of this work is to proactively expose a critical blind spot in current federated learning defenses, which predominantly focus on inspecting individual client updates. By demonstrating an "aggregation-emergent" attack, we aim to motivate researchers to develop more sophisticated defense mechanisms that incorporate post-aggregation validation. This paper serves as both a warning and a benchmark, providing a crucial tool for evaluating the robustness of future defense strategies and ultimately fostering the development of more secure and trustworthy federated learning systems.

## 9 REPRODUCIBILITY STATEMENT

To ensure the reproducibility of our research, we have provided a detailed description of our experimental setup in Section 5.1. This includes comprehensive information on the datasets (CIFAR-10), model architecture (ResNet-18), federated learning configurations, baseline attacks and defenses, and all relevant hyperparameter settings used in our evaluations. Furthermore, upon publication of this paper, we will make our source code, including the implementation of the CABA attack and the scripts used to generate our experimental results, publicly available to facilitate verification and further research in this area.

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

# APPENDIX

## A  THE LIMITATION OF CABA

Despite its demonstrated effectiveness, a primary limitation of CABA stems from its core design principle: the reliance on faithful aggregation. The attack's success is predicated on the precise and unaltered combination of the seemingly benign sub-model updates to reconstruct the malicious functionality in the global model. This dependency introduces a potential vulnerability. Specifically, CABA's efficacy can be compromised if the central server applies any pre-aggregation modifications to the incoming updates. Defense mechanisms or standard operational procedures such as gradient clipping, differential privacy (which involves adding noise), or model pruning, if applied individually to each update before the final summation, could disrupt the delicate, pre-calculated synergy among the collusive updates. Such modifications would likely alter the sub-models in unintended ways, thereby impeding the successful emergence of the backdoor in the aggregated global model and significantly degrading the ASR. Therefore, while CABA is robust against defenses that inspect updates in isolation, its performance is sensitive to server-side interventions that manipulate the updates themselves prior to aggregation.

Table 3: Performance comparison of various backdoor attacks against robust aggregation defenses in CIFAR-10 under the Non-IID setting.(Acc%/ASR%)

| Defense | Start Round | DBA | Chameleon | CoBA | 3Dfed | CABA |
|---|---|---|---|---|---|---|
| Nodefense | 400 | 91.05/89.41 | 91.55/39.91 | 91.25/99.56 | 83.14/92.78 | 91.63/90.37 |
| | 800 | 92.65/89.27 | 92.38/18.34 | 92.15/99.70 | 90.53/94.08 | 92.46/91.15 |
| | 1200 | 92.73/94.09 | 92.93/20.88 | 92.48/100.00 | 91.47/99.33 | 91.67/97.35 |
| Multi-Krum | 400 | 88.99/8.84 | 87.76/9.22 | 89.10/99.72 | 83.78/99.44 | 89.44/90.83 |
| | 800 | 90.92/8.81 | 90.70/9.81 | 90.77/99.53 | 87.08/97.77 | 89.36/97.55 |
| | 1200 | 92.32/9.69 | 91.81/8.87 | 92.53/100.00 | 92.24/98.18 | 92.21/91.63 |
| FLAME | 400 | 89.22/94.75 | 88.29/8.82 | 89.10/99.79 | 88.61/91.49 | 87.36/98.84 |
| | 800 | 91.06/95.08 | 91.30/9.38 | 91.80/99.99 | 89.81/9.6 | 90.41/93.16 |
| | 1200 | 92.21/93.58 | 91.70/9.59 | 92.28/99.93 | 90.0/99.08 | 92.66/92.12 |
| Foolsgold | 400 | 90.21/39.03 | 91.29/10.83 | 90.23/97.82 | 91.32/9.67 | 90.39/99.37 |
| | 800 | 92.10/9.84 | 92.07/12.80 | 91.45/98.33 | 91.95/9.81 | 91.49/94.53 |
| | 1200 | 92.71/10.35 | 92.73/15.68 | 92.66/99.99 | 92.5/94.94 | 92.61/90.74 |
| RFLBAT | 400 | 90.55/77.40 | 91.37/40.73 | 91.84/99.94 | 90.33/97.83 | 91.29/91.36 |
| | 800 | 92.14/94.00 | 92.37/53.93 | 91.92/99.95 | 75.44/6.63 | 91.78/97.42 |
| | 1200 | 92.74/94.82 | 92.87/57.08 | 92.62/99.99 | 92.01/93.63 | 92.32/93.74 |
| Deepsight | 400 | 90.79/63.88 | 91.45/9.64 | 91.89/60.53 | fail | 92.57/33.89 |
| | 800 | 91.20/58.77 | 91.97/9.44 | 92.09/14.25 | fail | 92.60/35.32 |
| | 1200 | 92.56/73.36 | 92.89/9.77 | 92.74/99.47 | fail | 92.79/32.96 |
| B.Indicator | 400 | 87.94/19.18 | 88.69/9.00 | 89.17/99.89 | 89.32/8.64 | 91.59/91.95 |
| | 800 | 89.13/20.32 | 89.93/8.40 | 91.59/99.86 | 85.55/56.87 | 91.48/91.90 |
| | 1200 | 92.24/83.94 | 92.14/9.46 | 92.42/99.62 | 92.60/17.54 | 92.19/93.66 |

Table 4: Experiment Parameters

| Parameter | Value | Parameter | Value |
|---|---|---|---|
| benign_lr | 0.1 | poisoned_dirichlet_alpha | 0.9 |
| benign_lr_gamma | 0.1 | poisoned_rate | 5/64 |
| benign_momentum | 0.9 | backdoor_lr | 0.05 |
| benign_retrain_no_times | 5 | backdoor_momentum | 0.9 |
| benign_weight_decay | 0.0005 | backdoor_original_class | 1 |
| indicator_threshold | 85 | backdoor_label_swap | 2 |
| deepsight_num_channel | 3 | backdoor_retrain_no_times | 5 |
| ood_data_batch_size | 64 | backdoor_weight_decay | 0.0005 |
| ood_data_sample_lens | 800 | batch_size | 64 |
| ood_data_source | CIFAR100 | tuned_trigger_round | 1 |
| poisoned_data_split_method | dirichlet | $\beta$ | 0.01 |
| $\alpha$ | 0.8 | $\gamma$ | 0.001 |

# B    EXPERIMENTAL DETAILS

For our experiments, we established a comprehensive set of parameters to ensure reproducibility and rigor, as detailed in Table 4. This table outlines the hyperparameters for both the benign and backdoor model training processes. Key settings include the learning rates (benign_lr = 0.1, backdoor_lr = 0.05), momentum (0.9 for both), and weight decay (0.0005 for both). The configuration also specifies parameters for our data handling methodology, such as utilizing CIFAR-100 as the out-of-distribution (ood) data source and employing a Dirichlet distribution with poisoned_dirichlet_alpha=0.9 for partitioning the poisoned data. Crucial coefficients for our proposed method are set as $\alpha$=0.8, $\beta$=0.01, and $\gamma$=0.001. Furthermore, the specifics of the backdoor attacks are defined in Table 5, which enumerates the four distinct trigger patterns by their precise pixel coordinates.

Table 5: Backdoor Trigger Patterns

| Pattern ID | Coordinates |
|---|---|
| Backdoor_pattern_0 | (0, 0), (0, 1), (0, 2), (0, 3), (0, 4), (0, 5) |
| Backdoor_pattern_1 | (0, 9), (0, 10), (0, 11), (0, 12), (0, 13), (0, 14) |
| Backdoor_pattern_2 | (4, 0), (4, 1), (4, 2), (4, 3), (4, 4), (4, 5) |
| Backdoor_pattern_3 | (4, 9), (4, 10), (4, 11), (4, 12), (4, 13), (4, 14) |

## C A CLOSE LOOT AT CABA

To illustrate the stealthiness of our proposed CABA method, we conduct a comparative analysis of the backdoor behavior exhibited by malicious models under different attack scenarios. Figure 5 presents the poisoned accuracy of four malicious models over successive communication rounds when subjected to the 3DFed defense mechanism, while Figure 6 shows the performance of malicious models constructed by CABA under the same conditions.

As depicted in Figure 5, the malicious models aggregated by 3DFed consistently maintain a very high backdoor accuracy, frequently approaching 100%. The performance curves of the four models are nearly identical, indicating that their malicious updates possess a high degree of similarity. This uniformity and consistently high success rate on the backdoor task make the malicious behavior conspicuous and easily detectable. The persistent, high-accuracy pattern serves as a clear signature of the attack, which can be leveraged by defense mechanisms.

In stark contrast, the behavior of malicious models generated by the CABA framework, shown in Figure 6, is significantly more erratic and unpredictable. The backdoor accuracy of each model fluctuates dramatically across rounds, without maintaining a consistently high level. Furthermore, the performance curves for the different models diverge, showing no discernible correlation or pattern. This volatility suggests that CABA does not rely on continuously injecting a strong, uniform backdoor into a single model. Instead, the backdoor behavior characteristics are strategically distributed among all sub-models across different rounds. By avoiding a persistent and high-impact malicious signature in any individual model update, CABA effectively conceals the attack, thereby achieving superior stealthiness and posing a greater challenge to existing defense strategies.

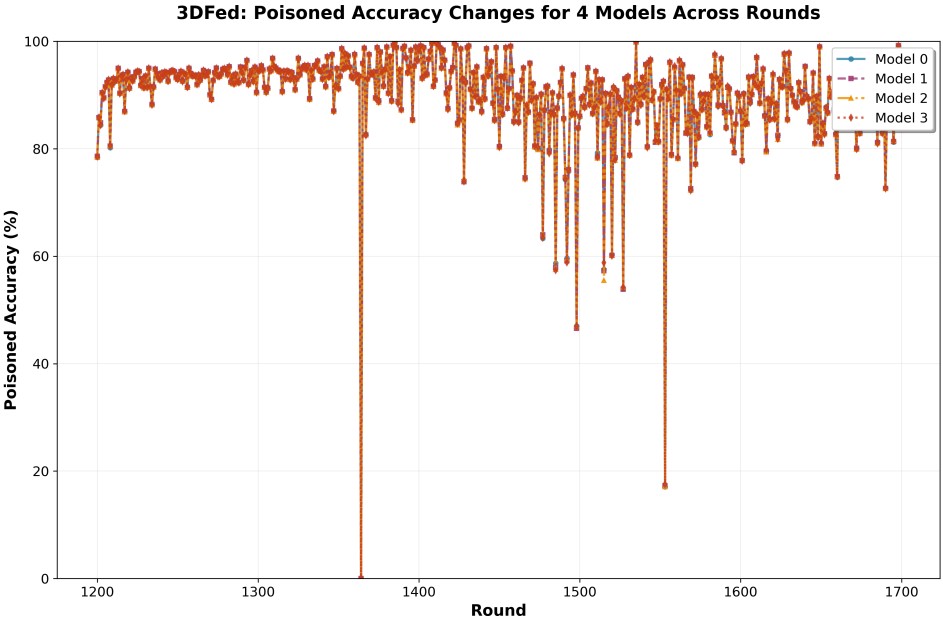

Figure 5: The backdoor accuracy of the malicious models uploaded by 3DFed varies with the number of communication rounds.

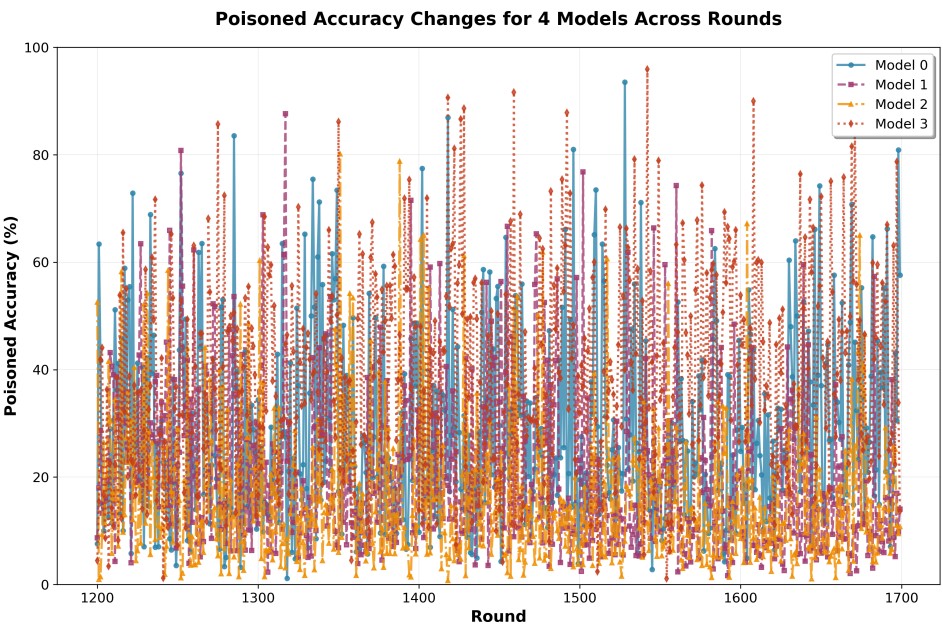

Figure 6: The backdoor accuracy of the malicious models uploaded by CABA varies with the number of communication rounds.

