# OpenReview forum: "CABA: A Collusive Aggregation-Emergent Backdoor Attack in Federated Learning"
_ICLR.cc/2026/Conference — ICLR 2026 Conference Withdrawn Submission_

### Official Review · Reviewer_7Dcr · 2025-10-17

**Soundness:** 2
**Presentation:** 2
**Contribution:** 2
**Rating:** 4
**Confidence:** 4

**Summary:**

This paper proposes CABA (Collusive Aggregation-based Backdoor Attack), a novel “aggregation-emergent” attack in federated learning (FL). Unlike previous collusive attacks where each malicious client independently embeds a full backdoor, CABA distributes the backdoor functionality across several malicious clients so that each individual update appears benign. The backdoor only emerges after aggregation, making it hard to detect via standard defenses. The authors conduct experiments on CIFAR-10 using ResNet-18 and evaluate against several state-of-the-art defenses (e.g., Multi-Krum, FLAME, FoolsGold, DeepSight, BackdoorIndicator), showing that CABA achieves high attack success rates and defense bypass capability. The paper claims this exposes a blind spot in existing FL defense mechanisms and calls for post-aggregation validation methods.

**Strengths:**

- The idea of an aggregation-emergent backdoor, which activates only after global model aggregation, represents a meaningful conceptual advancement beyond prior collusive paradigms.
- The paper compares against several strong baseline attacks (DBA, CoBA, 3DFed, Chameleon) and defenses, demonstrating empirical superiority across multiple FL settings (IID/Non-IID).
- The proposed framework is well-organized into four phases (Data Redistribution, Benign Pre-training, Joint Trigger Tuning, and Joint Backdoor Training), offering clarity in design and reasoning.

**Weaknesses:**

- The evaluation omits more recent or stronger defenses such as AlignIns (CVPR 2025). Without this, the claimed robustness may not generalize to the current defense landscape.
- The attack presumes that the same subset of malicious clients is always selected each round and that full synchronization between them is possible. This is unrealistic in real FL settings.
- Experiments only conducted on CIFAR-10 with ResNet-18, lacking evidence of generality across more datasets and models.
- The framework involves multiple key hyperparameters (α, β, γ) that balance stealth, diversity, and backdoor strength. However, there is no reported study on their influence or stability.
- CABA requires multi-client joint training, which could substantially increase communication and memory overhead. The paper lacks any runtime or memory cost analysis.
- The method is largely heuristic. No theoretical justification or formal analysis is given to explain why distributed sub-model interactions reliably produce an emergent backdoor after aggregation.
- No released code – Reproducibility cannot be verified.
- Figure 4 analyzes data distribution and attacker proportion but does not clearly show CABA’s ablation results.

**Questions:**

- How would CABA perform under AlignIns (CVPR 2025) or other feature-space alignment defenses?
- Could the authors provide runtime, GPU memory, and communication overhead comparisons relative to baseline attacks, and discuss whether the multi-client joint optimization scales to larger backbones (e.g., ViT-B/16)?
- The CABA loss combines multiple coefficients (α, β, γ). How sensitive are attack success and stealth to these parameters? Can the attack remain effective without precise tuning?

---

### Official Review · Reviewer_LHWc · 2025-10-30

**Soundness:** 2
**Presentation:** 1
**Contribution:** 2
**Rating:** 2
**Confidence:** 4

**Summary:**

The paper proposes CABA, a collusive backdoor attack in which multiple compromised clients jointly craft updates that appear benign individually, while the backdoor is implanted only after server aggregation of these updates. The pipeline comprises: (1) attacker-side data redistribution to induce diversity; (2) benign pre-training; (3) joint trigger tuning; and (4) joint backdoor training with a composite loss that embeds the backdoor signal in the aggregated model while regularizing each sub-model to remain inconspicuous. Experiments (e.g., CIFAR-10 / ResNet-18) report high attack success rate (ASR) and bypass of several server-side defenses.

**Strengths:**

- This attack studies more angle and exploits vulnerability when multiple clients are compromised, compared with previous attacks.
- The attack explicitly considers stealthiness, which is crucial for the security of federated learning and makes the threat model more realistic.
- The evaluation includes multiple robust-aggregation and behavior/property-based defenses, strengthening the contribution of this work.

**Weaknesses:**

- The presentation needs a strong revision. From the current version, it is unclear how the attack is conducted, and the Joint Trigger Tuning and Joint Backdoor Training sections miss many important details. For example:
  1. Based on Objective (4), the trigger is optimized such that the model still classifies triggered samples correctly. If that is the case, how does the adversary finally achieve the backdoor objective?
  2. What are $X_k$ and $Y_k$ in Eqn. (5)? What is the relationship between Eqn. (4) and Eqn. (5)?
  3. It is unclear how each compromised client trains its poisoned model and trigger, how local triggers are combined, and how these clients communicate.
  4. The novelty appears to involve splitting the adversary’s dataset into multiple shards to create several local models, but this is vague; the paper does not clearly explain how the attack is actually performed.

 - The evaluation is limited: there is no ablation on malicious dataset size, no exploration of non-IID hyperparameters, no tests with different aggregation algorithms or model architectures, and no examples of the final optimized triggers.
- All evaluated defenses are from before 2022.
- The captions of tables and figures are not informative. For example, Figure 2 does not highlight the differences between this method and existing FL attacks, and its caption lacks useful explanation.
- A systematic comparison against existing attacks and clearer representations (figures and tables) are recommended to better illustrate the novelty of this work.

**Questions:**

1. Define $X_k, Y_k$ vs. $\hat{X}_k, \hat{Y}_k$; clarify how these sets are sampled per step and how they differ from $X_b, Y_b$ used for the aggregated forward pass.
2. Do you alternate Eq. (4) and Eq. (5)? If so, what is the cadence, and how sensitive is ASR to this schedule and the $\alpha, \beta, \gamma$ parameters?
3. How does each malicious client compute gradient and how do they share gradients formally, and what is the per-round communication overhead among malicious clients?
4. Quantify ASR under per-update clipping (different norms), Gaussian noise (DP), or more complicated aggregators such as FedProx, Scaffold.

---

### Official Review · Reviewer_P4Gf · 2025-10-31

**Soundness:** 2
**Presentation:** 3
**Contribution:** 2
**Rating:** 4
**Confidence:** 4

**Summary:**

The paper introduces CABA (Collusive Aggregation-based Backdoor Attack), a new class of collusive attacks in federated learning (FL) where the backdoor behavior appears only after model aggregation — not in any single client’s update. Each malicious client uploads a harmless-looking “fragment” of the backdoor, evading detection by defenses that examine individual updates. CABA breaks traditional collusive attacks (e.g., DBA, CoBA, 3DFed) that involve training a full backdoored model by distributing the backdoor embedding process.
This paper benchmarks empirically on seven types of defenses and achieves better overall ASR performance while maintaining accuracy.

**Strengths:**

- This paper proposes a way to distribute the backdoor embedding process instead of using classical ways of training a full backdoored one.
- This paper is well structured with detailed definitions and respective theoretical framework.

**Weaknesses:**

- The trigger in this work is designed within the joint trigger tuning process. However, it is still unclear what the trigger is like and how the trigger is simultaneously activated across clients.
- Although the authors conduct extensive experiments to demonstrate the effectiveness of the CABA attack under various scenarios, including different defense and attack methods, the datasets and models used remain limited to only CIFAR-10 and ResNet-18.
- The proposed method heavily depends on the assumption that the FL dataset must be accessible and redistributed before launching the attack. However, this scenario may not be practical in real-world situations, as most FL setups typically do not permit access to client-side data.

**Questions:**

- Can you demonstrate the trigger more clearly with a visualization of how a trigger is optimized?
- Also, what are the differences between untuned triggers and tuned triggers in this work?
- Can you provide more experiments on other datasets (e.g., CIFAR-100, ImageNet) and with other models to show that the proposed method is scalable?
- Can you point out some scenarios, like CABA, where multiple clients might need to be triggered simultaneously and updated synchronously?

---

### Official Review · Reviewer_cswz · 2025-10-31

**Soundness:** 2
**Presentation:** 2
**Contribution:** 2
**Rating:** 2
**Confidence:** 5

**Summary:**

The paper presents a backdoor attack framework in Federated Learning (FL) that leverages the collaboration of multiple malicious clients to bypass existing defenses. The authors argue that current collusive attacks often have each malicious client independently training a full backdoor model, making their updates easily detectable by advanced defense mechanisms.

To address this, CABA (Collusive Aggregation-Based Backdoor Attack) introduces an *aggregation-emergent* paradigm, where the backdoor functionality materializes only during the server’s aggregation phase. Individual malicious clients contribute seemingly benign, fragmented updates that collectively induce a backdoor in the aggregated global model. The four-phase design—Data Redistribution, Benign Model Pre-training, Joint Trigger Tuning, and Joint Backdoor Training—ensures that no single malicious update exhibits a complete backdoor signature.

Experiments on CIFAR-10 demonstrate that CABA successfully circumvents six defense mechanisms, achieving both high attack success rates (ASR) and strong stealthiness compared to existing collusive and single-client attacks. The work highlights a significant blind spot in current defense paradigms that primarily focus on inspecting individual client updates, underscoring the need for post-aggregation validation.

**Strengths:**

1. Exposes a critical vulnerability in current defense mechanisms that only analyze individual client updates.
2. Introduces an innovative “aggregation-emergent” backdoor design, where the malicious effect arises solely after model aggregation—making individual updates appear benign.
3. The four-phase framework is clearly structured and effectively addresses challenges related to model diversity, trigger optimization, and update stealthiness.

**Weaknesses:**

- As discussed in Appendix A, CABA’s success relies heavily on "faithful aggregation," meaning its effectiveness degrades if the central server applies *any* pre-aggregation modifications (e.g., gradient clipping, differential privacy, or pruning). These are common in modern FL deployments, which significantly limits CABA’s real-world applicability.
- The paper describes the trigger as a "pixel pattern" and provides coordinates in Table 5, but a visual example of trigger-embedded images would help assess perceptibility. Moreover, while Figure 6 presents erratic sub-model backdoor accuracies, a qualitative visualization showing how the *aggregated model* consistently activates the backdoor would better support the "aggregation-emergent" claim.
- CABA assumes perfect coordination among malicious clients and exact aggregation of "benign-looking fragments." In practical FL systems with network noise, varying client computation, or deviations in aggregation logic, this synergy may break down, weakening the emergent backdoor effect.
- The multi-phase pipeline—requiring data redistribution, joint training, and synchronized optimization across malicious clients—introduces substantial coordination and communication overhead. This level of orchestration may be impractical for large-scale or geographically dispersed adversaries. The paper should quantify this cost and discuss its feasibility relative to single-client attacks.
- CABA requires all malicious clients to submit their updates simultaneously each round to sustain the emergent backdoor. The paper does not examine cases where some attackers fail to participate or submit benign models, which would clarify the sensitivity of the attack to partial collaboration.
- Non-IID settings are more representative of real-world FL but are relegated to the appendix. These results should appear in the main paper for better visibility and evaluation.
- Table 1 shows fluctuating backdoor accuracy for CoBA under Multi-Krum and FLAME (e.g., low accuracy at round 1200 but high at 400/800). The authors should explain this behavior and specify the attack schedule—continuous or periodic—as well as justify using 20 clients per round instead of the more common 10-client setup in prior work.
- The failure of Deepsight in Table 3 should be explicitly explained to clarify whether it stems from detection errors, instability, or incompatibility with the aggregation-emergent mechanism.
- Experiments are confined to CIFAR-10. Evaluating CABA on more complex datasets (e.g., CIFAR-100, ImageNet) and grayscale datasets (e.g., MNIST, FashionMNIST) would better demonstrate generalizability across different data modalities and tasks.

**Questions:**

Please address the aforementioned weaknesses. Specifically, clarify CABA’s robustness to aggregation modifications, discuss coordination overhead, provide trigger visualizations, explain anomalous results, and expand experiments to more datasets.

---

### Note · Authors · 2025-11-26

I have read and agree with the venue's withdrawal policy on behalf of myself and my co-authors.